# Associations of genetic variants for educational success with risk and time preferences vary by childhood environment
**Chris Dawson** ✉

Prior research shows that individuals with higher cognitive ability tend to be more patient and less risk-averse, while childhood environments also exert a strong influence on the development of these preferences. This raises the question of whether associations between cognition and economic preferences are consistent across early-life contexts. I test this using incentivized experimental ($N =$ 624) and survey ($N =$ 5,881; 11,521 person-wave observations) measures of risk and time preferences, detailed indicators of childhood environments, and a polygenic score for educational attainment—capturing genetic variances associated with cognitive and non-cognitive traits relevant to educational success. I find that genetic variance related to educational success is associated with lower risk aversion and greater patience, but only among individuals raised in more advantaged childhood environments. Among those who experienced childhood adversity, this genetic variance instead predicts greater risk aversion, and its association with patience is substantially attenuated. These patterns suggest that early adversity may canalize, constrain, or redirect the developmental expression of cognitive-relevant genetic variances in ways that are adaptive to context. Causal research is needed to ascertain if such environmentally contingent expression of genetic variances can reinforce patterns of social immobility.

Economic preferences—such as patience and risk-taking—play a fundamental role in shaping individual behavior and long-term socioeconomic outcomes. For example, more patient individuals are less likely to engage in criminal activity[1], and tend to achieve higher levels of educational attainment, occupational status, income, and wealth[2–4]. Similarly, individual differences in risk preferences are associated with labor market performance, investment behavior, health outcomes, and addictive behaviors[5–8]. Understanding the development of these preferences is therefore central to research in economics, psychology, and the social sciences more broadly.

A large body of research identifies two powerful predictors of economic preferences: early-life environments and cognitive abilities. Empirical work has shown that individuals with higher cognitive ability tend to exhibit greater patience and lower risk aversion[9–11], and that higher cognitive ability is linked to the same socioeconomic advantages as patience and lower risk aversion—including higher income, occupational attainment, and wealth accumulation[12–17]. Several mechanisms explain why cognition is related to decision-making under risk and over time. Cognitive constraints can lead to

narrow bracketing[18–21], whereby individuals fail to recognize how risky or intertemporal choices relate to broader consequences, such as lifetime wealth[22,23], thereby producing more risk-averse and myopic decisions. A further channel is provided by the two-system approach to decision-making from the 'heuristics and biases' literature[24], which emphasizes the interplay between emotions and cognition in decision-making. Here, risk aversion and myopic preferences are thought to be universal properties of System 1, the emotional system that operates with little or no effort, and no sense of voluntary control[25]. Those low in cognitive resources are typically regarded as less able to override the emotional system and engage the deliberative, logical and analytical system, System 2, where reasoning is risk neutral and patient[26–30].

Early-life environments also have a strong association with economic preferences[31–35]. However, theoretical perspectives diverge. Life-history theory[36] argues that childhood stress and resource scarcity promote a "fast" strategy characterized by impatience and greater risk-taking, whereas stable, resource-rich settings encourage a "slow" strategy of patience and lower

School of Management, University of Bath, Claverton Down, Bath, UK. ✉e-mail: c.g.dawson@bath.ac.uk

risk-taking[37–40]. In contrast, the uncertainty-management perspective sees early adversity as a signal of future danger, shaping preferences that help manage uncertainty, such as impatience and risk aversion[41,42]. Empirically, childhood stress reliably predicts higher impatience, while evidence on risk taking is mixed—some studies report elevated risk-taking[40,43], whereas others find greater risk aversion[42].

Despite extensive work in both domains, these two determinants—cognition and childhood environment—are typically studied independently. As a result, we still lack a clear understanding of how early environments condition the association between cognitive factors and economic preferences. This article addresses this gap by leveraging cognitive-related genetic data—specifically, a polygenic score for educational attainment—to test whether associations with patience and risk-taking differ for individuals raised in advantaged versus disadvantaged childhood environments. Such evidence sheds light on whether genetic variations operate uniformly or whether their behavioral expression depends on the developmental environments in which individuals grow up.

Theoretical frameworks provide clear reasons to expect such moderation. The Experiential Canalization Framework[44,45] proposes that early-life adversity can impose pervasive constraints that limit the developmental processes and pathways that are thought to link genetic variation to behavior. Heightened emotional reactivity, chronic stress, negative affect, and reduced emotional regulatory control—well-established consequences of adverse childhood conditions[46–52]—shift decision-making toward intuitive, affect-driven processes (System 1) and away from deliberative reasoning (System 2)[29,53–58]. In a similar way, negative affect and anxiety can influence decision-making by inducing a generalized overestimation of risks[59,60] and an attentional bias toward threatening information[61]—potentially attenuating or even blocking the association between genetic variation and behavioral expressions of cognitive ability.

Complementing the Experiential Canalization Framework perspective, Adaptive Calibration Models[62] propose that early environments do not merely suppress the expression of genetic variations but redirect them in contextually adaptive ways. In this framework, biological systems—including stress reactivity and attentional processes—are calibrated by early-life cues to prepare individuals for the kinds of environments they are likely to encounter. Accordingly, in stable, resource-rich settings, cognitive resources may be channeled toward long-term planning, exploration, and opportunity recognition, whereas in harsh or unpredictable contexts, those same resources may be oriented toward vigilance, uncertainty management, and short-term security[63]. Evidence that childhood adversity can heighten cue detection and increase physiological sensitivity to environmental challenge—especially among individuals with greater processing capacities[64,65]—supports this interpretation. This aligns with research showing that early-life stress can lead to the specialization of certain cognitive functions[66,67]. Moreover, cross-national evidence indicates that higher cognitive ability predicts greater patience and risk-taking in high-income contexts but lower patience and risk-taking in low-income contexts[68], consistent with the idea that ecological conditions structure how cognitive resources are deployed.

To test these propositions, I draw on a dataset of 624 individuals that includes incentivized experimental measures of time and risk preferences, detailed indicators of early-life environments, and molecular genetic data, complemented by a larger sample of 5881 individuals (11,521 person-wave observations) with a validated survey-based measure of time preferences. Across both samples, there are rich indicators of childhood socioeconomic conditions—including parental human capital, material resources, and household (in)stability—as well as a polygenic score for educational attainment, which captures genetic variation related to cognitive and non-cognitive traits relevant to educational success[69–72]. These data allow us to provide insight into the pathways through which genetics and early experience jointly relate to decision-making, and, more broadly, to understand the possible origins of economic behavior and the factors associated with socioeconomic attainment and intergenerational inequality[73,74].

## Methods

### Inclusion and ethics statement

This study is based on the analysis of anonymized secondary data from the English Longitudinal Study of Ageing (ELSA) available from the U.K. Data Service. Information on the ethical approval received for each wave of ELSA can be found at: https://www.elsa-project.ac.uk/ethical-approval. This study (application reference number: 10956-12470) received a favorable opinion through the University of Bath's Social Science Research Ethics Committee's review process.

### Open science and data

This study was not preregistered. The data that support the findings of this study are publicly available from the U.K. Data Archive. All Stata analysis scripts to replicate the results are available on OSF: osf.io/f8teh.

### Participants

I use data from the English Longitudinal Study of Ageing (ELSA), Waves 0–10 (1998–2023)[75]. ELSA is a nationally representative panel study of adults aged 50 and over in England, originally drawn from the Health Survey for England. The study also includes some respondents under age 50—specifically cohabiting partners and spouses of core sample members, as well as a younger partner cohort included in several early waves. Participants are re-interviewed approximately every two years, with refresher samples introduced in Waves 3, 4, 6, 7, 9, and 10 to maintain representativeness of the target population.

In Wave 5, 1063 respondents aged 50–75 completed an incentivized experimental module designed to measure risk and time preferences. Participants in the experimental module were paid a participation fee of £10. The choice tasks in the experimental module involved real (but small) payoffs, and at the end of the module, one task was randomly selected, and participants received the amount corresponding to their choice for that task. Participants could not lose more than £5 from their initial £10 participation fee and the expected payment was approximately £35.

ELSA also provides genotyped data for 7412 participants, with polygenic scores (PGSs) available for several behavioral phenotypes, including educational attainment. I combine the Wave 5 experimental data with the genotyped data—restricting the analytic sample to participants of European genetic ancestry, consistent with the genome-wide association studies used to derive the polygenic scores—and the life-history interview module from Wave 3, which captures information about key life events, including childhood environment. This yields an analytic sample, which I refer to as the experimental sample, of 624 individuals who had valid polygenic scores, completed the incentivized experimental module, and completed the life-history interview. The mean age of the experimental sample was 64.18 years (SD = 5.61, range 50–75). Sex was self-reported in response to a prompt asking participants to indicate their biological sex (Female or Male). In this sample, 335 participants (53.7%) reported female, and 289 participants (46.3%) reported male.

Additionally, I use a survey-based measure of time preferences from Waves 1 and 2, resulting in a separate analytic sample, the survey sample, comprising 11,521 person-wave observations from 5881 individuals. The mean age of the survey sample was 64.46 years (SD = 9.43, range 31–90). Sex was self-reported in the same manner as in the experimental sample. In this survey sample, 3209 participants (54.6%) reported female and 2672 participants (45.4%) reported male.

Supplementary Table 1 provides an overview of the sample characteristics for the key variables used in the subsequent analysis. All ELSA participants provided informed consent at the time of their interview, including additional consent for biomarker and genetic data collection, where applicable.

### Childhood disadvantage

I take the approach in Ronda et al.[76] and consider childhood disadvantage over four dimensions. The first dimension is human capital disadvantage, which is measured as having neither parent with education above the

compulsory level, which at the time (1918–1947) meant children had to stay in school until the age of 14 (49.84% of the experimental sample; 41.71% of the survey sample). The second dimension measured family resources and is measured using a question on the main carer's occupation when the respondent was aged 14. The classifications include armed forces; manager or senior official; running own business; professional or technical; administrative, clerical or secretarial; skilled trade; caring, leisure, travel or personal services; sales or customer service; plant, process or machine drivers; other jobs; something else; casual jobs; retired; unemployed; and lastly, sick/disabled. Family resource disadvantage was main carer's occupation falling into either of the following categories: plant, process or machine drivers; other jobs; something else; casual jobs; unemployed; and lastly, sick/disabled (27.56% of the experimental sample; 33.14% of the survey sample). The third dimension is another measure of family resources disadvantaged, measured as growing up in a home with no central heating or hot water (23.72% of the experimental sample; 28.72% of the survey sample). The fourth dimension is a measure of family instability. I measured instability as parents being permanently separated or divorced before the respondent was aged 16, or having not lived with both natural parents for most of their childhood (15.22% of the experimental sample; 16.31% of the survey sample). In the experimental (survey) sample, 27.72% (28.45%) respondents experienced no dimension of disadvantage, 37.18% (35.88%) experienced one dimension of disadvantage, 27.08% (24.55) experienced two dimensions of disadvantage, 7.05% (9.59%) experienced three dimensions of disadvantage, and lastly, 0.96% (1.53%) experienced all four dimensions of disadvantage. Following Ronda et al.[76], I categorize disadvantage as those respondents who experienced two or more dimensions of disadvantage, which corresponds to 35.10% (35.67%) of the experimental (survey) sample. It should also be noted that for the experimental (survey) sample, 29.01% (25.06) respondents had a missing response to one of the four disadvantage dimensions, with 1.76% (16.51%) of respondents having a missing response to two of the four disadvantage dimensions. To maintain statistical power, I treat missing values as zero.

I show that the main results (Supplementary Tables 2 and 3) are almost identical when restricting the sample to those respondents who gave valid responses to each of the four dimensions of disadvantage (Supplementary Tables 4 and 5) and using the missing indicator method, whereby I include a dichotomous control variable for missingness (Supplementary Tables 6 and 7). I also consider alternative specifications of childhood disadvantage, including each dimension individually (Supplementary Tables 8 and 9) and the number of dimensions experienced during childhood (Supplementary Tables 10 and 11).

## Polygenic score (PGS) for educational attainment (EA)

Polygenic scores (PGSs) represent genetic variation statistically associated with a given phenotype. They are calculated as the sum of genome-wide genotypes—i.e., the combination of alleles across many loci—weighted by effect sizes (e.g., beta coefficients) derived from genome-wide association study (GWAS) summary statistics. GWAS test hundreds of thousands of genetic variants, typically single nucleotide polymorphisms (SNPs), to identify those statistically associated with a specific phenotype; in this case, educational attainment (EA). Because most phenotypes are polygenic, involving many SNPs of small effect, an individual's polygenic score is calculated as a weighted sum of their SNP genotypes:

$$PGS_i^{EA} = \sum_{j=1}^{j} W_j^{EA} G_{ij} \qquad (1)$$

where $PGS_i^{EA}$ is the educational attainment polygenic score for individual $i$, $G_{ij}$ represents the genotype of individual $i$ at SNP $j$, and $W_j^{EA}$ is the GWAS-derived effect size (beta) for SNP $j$.

ELSA used a single $p$-value threshold of 1 (i.e., including all SNPs), as prior research has shown that such PGSs either explain the most phenotypic variance or perform comparably to scores constructed using more restrictive thresholds[77]. The PGS for educational attainment used here is based on GWAS summary statistics from Lee et al.[70]; additional details on PGS construction in ELSA are provided elsewhere[78].

The PGS EA captures genetic associations with a broad set of cognitive and neurobiological traits, including brain volume, learning, neural function, cognitive development, and regions of the brain associated with language, memory, visual recognition, and higher-order processing, as well as the 'g' factor of intelligence[69–71,79,80]. At the same time, research shows that the PGS EA is also associated with non-cognitive traits relevant to educational success, including personality dimensions such as Neuroticism and Openness to Experience[72,81].

In the current data, individuals with higher PGS EA tend to have higher phenotypic educational attainment in both the experimental and survey samples (experimental: $r(622) = 0.195$, $p < 0.001$, 95% CI [0.119, 0.270]; survey: $r(11,519) = 0.196$, $p < 0.001$, 95% CI [0.178, 0.213]), as well as higher phenotypic cognitive ability (experimental: $r(622) = 0.190$, $p < 0.001$, 95% CI [0.113, 0.265]; survey: $r(11,519) = 0.145$, $p < 0.001$, 95% CI [0.127, 0.162]). These correlations closely match those reported in Lee et al.[70] and provide face validity for the polygenic measures used here. Further description of the cognitive phenotypes is provided in Supplementary Note 1. In addition, detailed methods and analyses of the relationship between PGS EA and both educational attainment and cognitive ability—including tests across levels of childhood disadvantage—are provided in Supplementary Note 2, together with Supplementary Table 24 and Supplementary Fig. 1.

## Risk preferences

In Wave 5, as part of the incentivized experimental module designed to measure risk and time preferences, participants responded to the incentive-compatible task by Binswanger[82,83] and Eckel and Grossman[84] (B-EG). The B–EG procedure required respondents to choose one of six lotteries, each offering a low and high payoff with equal probability. These lotteries are shown in Table 1. Assuming Expected Utility Theory (EUT) and Constant Relative Risk Aversion (CRRA)—and assuming that the experimental prizes are integrated into a background income equal to zero—the coefficient of relative risk aversion associated to each choice corresponds to the intervals reported in the rightmost column of Table 1. Within this CRRA functional form, a coefficient equal to zero implies risk-neutrality, coefficients above zero denote risk aversion, and coefficients below zero imply risk-seeking behavior. Responses to the B-EG task allow us to infer the upper and lower bounds of each participant's risk-aversion coefficient, and I use these bounds directly in the empirical analysis.

Respondents were also asked to complete an adapted version of the B-EG procedure that includes losses. Indeed, loss aversion[85]—the tendency to feel more pain when experiencing losses than pleasure from equal gains—is the primary driver of risk-taking behavior in the context of small-stakes gambles[86]. These lotteries are shown in Table 2, alongside the expected values and standard deviations. I utilize the full distribution of responses from the adapted B-EG procedure in the empirical analysis, as well as a

**Table 1 | B-EG lottery probabilities, expected value, standard deviation and CRRA range**

| Lottery | Payoff | | | | CRRA ranges | |
| | Low | High | Expected value | Std. Dev. | Lower bound | Upper bound |
|---|---|---|---|---|---|---|
| A | 28 | 28 | 28 | 0 | 3.46 | ∞ |
| B | 24 | 36 | 30 | 8.5 | 1.16 | 3.46 |
| C | 20 | 44 | 32 | 17 | 0.71 | 1.16 |
| D | 16 | 52 | 34 | 25.5 | 0.499 | 0.71 |
| E | 12 | 60 | 36 | 33.9 | 0 | 0.499 |
| F | 2 | 70 | 36 | 48.1 | −∞ | 0 |

## Table 2 | Loss aversion lottery probabilities, expected value and standard deviation

| Lottery | Payoff | | Expected value | Std. Dev. |
|---|---|---|---|---|
| | Low | High | | |
| A | 10 | 10 | 10 | 0 |
| B | 6 | 18 | 12 | 8.5 |
| C | 2 | 22 | 12 | 14.1 |
| D | −2 | 28 | 13 | 21.2 |
| E | −4 | 35 | 15.5 | 27.6 |
| F | −5 | 38 | 16.5 | 30.4 |

## Table 3 | Payoff table for time preference tasks

| Payoff Alternative | Payment Option A (Two Weeks) | Payment Option B (One Month) | Payment Option B* (Two Months) | Switching Point – Weekly Discount Rate | |
|---|---|---|---|---|---|
| | | | | Lower bound | Upper bound |
| *Panel A* | | | | | |
| 1 | £25 | £26 | | −∞ | 0.0163 |
| 2 | £25 | £28 | | 0.0163 | 0.0477 |
| 3 | £25 | £30 | | 0.0477 | 0.0779 |
| 4 | £25 | £32 | | 0.0779 | 0.1069 |
| 5 | £25 | £35 | | 0.1069 | 0.1485 |
| 6 | £25 | £38 | | 0.1485 | 0.1880 |
| * | | | | 0.1880 | ∞ |
| *Panel B* | | | | | |
| 1 | £25 | | £26 | −∞ | 0.0059 |
| 2 | £25 | | £30 | 0.0059 | 0.0275 |
| 3 | £25 | | £35 | 0.0275 | 0.0514 |
| 4 | £25 | | £37 | 0.0514 | 0.0602 |
| 5 | £25 | | £40 | 0.0602 | 0.0726 |
| 6 | £25 | | £45 | 0.0726 | 0.0915 |
| * | | | | 0.0915 | ∞ |

*Note.* These weekly discount rates assume exponential discounting, specifically, that the present value $PV$ of a future reward $FV$ is given by: $PV = FV/(1 + d)^t$, where $d$ is the discount rate and $t$ is the time delay in weeks. * Indicates the discount rate bounds for those participants who never switched, that is, those that always chose Payment Option A.

binary indicator which equals one if the respondent picked Lotteries A, B or C, and zero otherwise (i.e., Lotteries D, E or F, that contain losses).

### Time preferences

As part of the Wave 5 experimental session, following Harrison et al.[87] and Andersen et al.[88], participants completed a multiple price list (MPL) designed to elicit time preferences. Respondents chose between £25 in a "sooner" period or £25 + x in a "future" period *time t + T*, where x > 0. The "sooner" period was two weeks' time, and the "future" periods corresponded to 1 month and 2 months. Each participant therefore completed two MPLs. The full set of choice options is presented in Table 3.

I focus on the point at which a respondent switches from choosing the "sooner" option (Option A) to choosing the "future" option (Option B), which provides the upper and lower bounds of the respondent's implied discount rate. For example, if an individual selects Option A in Payoff Alternative 4 in Panel A of Table 3 but switches to Option B in Payoff Alternative 5, then—under exponential discounting—the respondent's weekly discount rate lies between 10.69% and 14.85%. Thus, responses to the MPL tasks allow us to infer interval-valued estimates of participants'

discount rates over both time horizons, and I use these bounds directly in my empirical analysis.

Some respondents exhibit extreme impatience or patience and never switch, choosing Option A (or Option B) in all alternatives. For these individuals, the inferred bounds on the weekly discount rate are less precise. A further subset of respondents make inconsistent choices—for example, switching from Option A to Option B and then back to Option A. In the MPL with a 1-month (2-month) future delay, 60 (47) respondents displayed such inconsistencies. For these cases, I code their first switching point as their implied interval and include in all analyses a dichotomous control variable equal to one for inconsistent respondents and zero otherwise.

I also supplemented my experimental measure of time preferences with a question on planning horizons from the main survey. Specifically, in Waves 1 and 2 respondents were asked: *"In deciding how much of your income to spend or save, people are likely to think about different financial planning periods. In planning your saving and spending, which of the following time periods is more important to you?"*. This question has been used in previous research as a marker of more general future-orientation, or time perspective in financial behavior[89,90] and is correlated with other established markers of time perspective, including smoking and body mass index[91]. Respondents can choose from seven possible answers, indicating planning timelines ranging from: '*does not plan/plans day to day*'; '*the next few weeks*'; '*the next few months*'; '*the next year*'; '*the next few years*'; '*the next 5–10 years*'; or '*longer than 10 years*'. In addition to analyzing the full distribution of responses to the spending horizon question, I construct a binary indicator equal to one if the respondent's reported horizon extends beyond the next year, and zero otherwise. This threshold aligns with standard distinctions in the strategic management and financial planning literature between short-term and long-term financial planning.

### Analytic strategy

I examine the relationship between PGS EA and economic preferences and compare the estimated genetic associations across childhood environments. This empirical strategy follows the standard between-family model for gene-by-environment interactions[76,92], specified as:

$$Y_{ig}^h = \alpha^h(g)PGS_{ig}^{EA} + X_{ig}b^h(g) + \epsilon_{ig}^h \quad (2)$$

where $Y_{ig}^h$ is the *h*th economic preferences (i.e., risk or time preference) for individual *i* in group *g* (i.e., 'no disadvantage' or 'disadvantage'), $PGS_{ig}^{EA}$ is the standardized polygenic score for educational attainment, and $X_{ig}$ is a vector of exogenous control variables including age (in linear and quadratic form), sex, and the first 10 principal components of the respondent's SNPs—which adjust for any ancestry differences in genetic structure. Adjusting for ancestry differences is important: if a particular SNP variant is more common in a specific ancestry group, an observed association between $PGS^{EA}$ and economic preferences may otherwise be more likely to reflect cultural norms shared by that group rather than genetic effects. Lastly, $\epsilon_{ig}^h$ is the usual random error component. This specification allows all parameters to differ across childhood environments, and my primary interest lies in the estimated genetic associations $\alpha^h$ and the comparison of the estimated genetic associations between groups *g*.

I estimated models using logistic regression for binary outcomes, ordered logistic regression for ordinal outcomes, and interval regression for outcomes that are interval-censored. For logistic and ordered logistic regression, I assumed independence of observations, no perfect multicollinearity, and linearity in the log-odds. The proportional odds assumption for ordered logistic models was tested and satisfied. Interval regression assumed independent, homoscedastic, and normally distributed latent errors. Residuals were visually inspected, and given the sample sizes, minor deviations are unlikely to affect inference.

In the standard between-family model for gene-by-environment interactions, PGS EA is often treated as plausibly exogenous, owing to Mendel's Law of Segregation, whereby individuals inherit genes randomly from their parents. Nevertheless, potential bias can arise from unmeasured

**Fig. 1 | Distribution of standardized PGS EA across childhood environments.** Panel (**A**) shows the kernel density distribution of the standardized polygenic score for educational attainment (PGS EA) in the experimental sample (N = 624). Panel (**B**) shows the corresponding distribution in the survey sample (N = 5881; 11,521 person-wave observations). In both panels, the black solid line represents respondents who experienced childhood disadvantage, and the blue dashed line represents respondents without childhood disadvantage. Density values are plotted on the vertical axis, and standardized PGS EA scores are plotted on the horizontal axis.

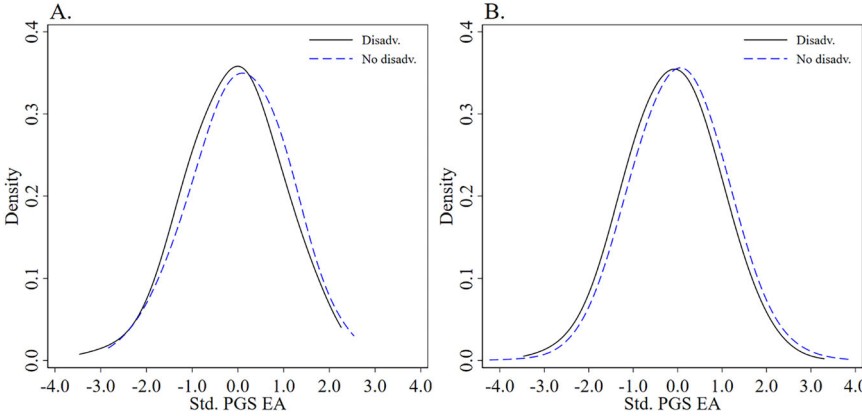

environmental influences, particularly those arising from genetic nurture. Simply put, individuals' genetic profiles are correlated with parental genes, which are also associated with features of the environments in which children are raised[93–96].

Figure 1 illustrates this point: the distribution of PGS EA differs across childhood environments. In the experimental sample, individuals from disadvantaged backgrounds have PGS EA values that are, on average, 0.097 SD below the overall mean. A two-sample $t$-test indicated that the mean PGS EA did not significantly differ between the groups, $t(622) = 1.77$, $p = 0.078$, $d = 0.148$, 95% CI [−0.016, 0.313]. A similar pattern appears in the survey sample. Here, individuals from disadvantaged backgrounds have PGS EA values 0.101 SD lower than the overall mean, and a two-sample $t$-test showed a statistically significant difference between groups, $t(11,519) = 8.10$, $p < 0.001$, $d = 0.157$, 95% CI [0.119, 0.196]. Such differences are expected: individuals inherit their genetic makeup from their parents, whose own genetic traits will be associated with their educational attainment, occupational status, and acquisition of resources. However, the correlation between PGS EA and the observed childhood disadvantage indicator is small in both samples (experimental: $r(622) = −0.071$, $p = 0.078$, 95% CI [−0.148, 0.008]; survey: $r(11,519) = −0.075$, $p < 0.001$, 95% CI [−0.093, −0.057]), indicating only a very weak correlation.

In this setting, particular attention is warranted for unobserved—rather than observed—differences in childhood environments. If other early-life factors correlated with respondents' PGS EA—such as parenting practices or parental time and resource investments—are also correlated with economic preferences, a between-family design cannot fully disentangle parental genetic variation associated with offspring outcomes from associations attributable to offspring genetic variation. This poses a potential challenge for identifying gene-by-environment interactions, as differences in the estimated associations between PGS EA and outcomes across groups may reflect unmeasured childhood environmental factors rather than differences in the strength of genetic associations. Accordingly, I conduct robustness checks using broad indicators of parenting styles as a diagnostic to assess whether parental behaviors correlated with PGS EA materially alter the estimated associations.

Additionally, PGS EA captures not only cognitive-related genetic associations[69–71,80] but also exhibits genetic overlap with personality domains such as Neuroticism and Openness to Experience[72]. To more precisely characterize the relationship between cognitive-related genetic associations and economic preferences, I assess whether including polygenic scores for the Big Five personality traits materially affects the results. To further isolate cognitive pathways, I also estimate models that substitute PGS EA with the polygenic score for IQ (PGS IQ) and with respondents' phenotypic cognitive ability. This approach provides a more direct assessment of the association between cognitive factors and economic preferences while remaining complementary to the main specification.

### Reporting summary
Further information on research design is available in the Nature Portfolio Reporting Summary linked to this article.

### Results
Table 4 reports the correlations between PGS EA and the preference measures separately for respondents who experienced childhood disadvantage and those who did not. These descriptive patterns suggest that the relationship between genetic variance associated with educational attainment and both risk and time preferences differs across early-life environments. To formally examine these differences, I estimate Eq. (2), with standardized PGS EA as the predictor, a binary indicator of early-life disadvantage as the moderator, and controls for age (in linear and quadratic form), sex, and the first 10 principal components of respondents' SNPs; all parameters are allowed to vary across the two childhood environment groups.

I begin with the experimental sample (N = 624): risk aversion in the B–EG task is analyzed using an interval regression, based on the upper and lower bounds of participants' implied risk-aversion coefficients (Table 1), with larger values corresponding to greater risk aversion. The main effect, that is, the estimated association for PGS EA and risk aversion for respondents without childhood disadvantage, was negative and statistically significant, $z = −3.00$, $p = 0.003$, $\beta = −0.368$, SE = 0.123, 95% CI [−0.609, −0.128]. The interaction effect with childhood disadvantage was positive and significant, indicating that the estimated association of PGS EA and risk aversion differs across the groups, $z = 3.61$, $p < 0.001$, $\beta = 0.799$, SE = 0.221, 95% CI [0.365, 1.233]. The implied association of PGS EA and risk aversion for the 'disadvantage' group was positive and statistically significant, $z = 2.33$, $p = 0.020$, $\beta = 0.431$, SE = 0.185, 95% CI [0.069, 0.793]. In short, higher PGS EA statistically predicts lower risk aversion among individuals without childhood disadvantage. In contrast, among those who experienced childhood disadvantage, higher PGS EA predicts greater risk aversion. The predicted risk aversion (Panel A, Fig. 2) spans from 1.602 ($z = 6.03$, $p < 0.001$, SE = 0.266, 95% CI [1.081, 2.123]) at high PGS EA ( + 2 SD from the mean), to 3.074 ($z = 10.41$, $p < 0.001$, SE = 0.295, 95% CI [2.495, 3.653]) at low PGS EA (−2 SD from the mean), in the 'no disadvantage' group, while the pattern reverses in the 'disadvantage' group, ranging from 3.425 ($z = 7.55$, $p < 0.001$, SE = 0.454, 95% CI [2.536, 4.315]) at high PGS EA to 1.702 ($z = 4.41$, $p < 0.001$, SE = 0.386, 95% CI [0.945, 2.458]) at low PGS EA.

Similar results are found when I analyze the risk aversion (LA) task using an ordered logistic regression. This measure is based on an adapted B–EG lottery that incorporates losses (Table 2) and is coded on a 6-point scale from 1 (most risk-seeking: Lottery F) to 6 (most risk-averse: Lottery A). The main effect shows that for respondents without childhood disadvantage, PGS EA significantly predicted lower risk aversion, $\beta = −0.350$, SE = 0.086, $z = −4.07$, $p < 0.001$, OR = 0.704, 95% CI [0.595, 0.834], with a significant positive interaction effect between PGS EA and childhood disadvantage, $\beta = 0.715$, SE = 0.173, $z = 4.13$, $p < 0.001$, OR = 2.045, 95% CI

**Table 4 | Correlations between PGS EA and economic preferences by childhood environment group**

| Group | Variable | r | p-value | 95% CI | n |
|---|---|---|---|---|---|
| Disadv. | Risk aversion (B-EG) | 0.097 | 0.155 | −0.037, 0.226 | 219 |
| No disadv. | Risk aversion (B-EG) | −0.142 | **0.004** | −0.236, −0.045 | 405 |
| Disadv. | Risk aversion (LA) | 0.086 | 0.206 | −0.047, 0.216 | 219 |
| No disadv. | Risk aversion (LA) | −0.212 | **<0.001** | −0.303, −0.117 | 405 |
| Disadv. | Risk aversion (LA-binary) | 0.030 | 0.657 | −0.103, 0.162 | 219 |
| No disadv. | Risk aversion (LA-binary) | −0.213 | **<0.001** | −0.304, −0.118 | 405 |
| Disadv. | Discount rate (1-month MPL) | −0.177 | **0.009** | −0.302, −0.045 | 219 |
| No disadv. | Discount rate (1-month MPL) | −0.139 | **0.005** | −0.233, −0.042 | 405 |
| Disadv. | Discount rate (2-month MPL) | −0.120 | 0.076 | −0.249, 0.012 | 219 |
| No disadv. | Discount rate (2-month MPL) | −0.174 | **<0.001** | −0.267, −0.078 | 405 |
| Disadv. | Planning horizon | 0.030 | 0.053 | −0.000, 0.061 | 4119 |
| No disadv. | Planning horizon | 0.095 | **<0.001** | 0.073, 0.118 | 7402 |
| Disadv. | Planning horizon (binary) | 0.015 | 0.331 | −0.015, 0.046 | 4119 |
| No disadv. | Planning horizon (binary) | 0.082 | **<0.001** | 0.059, 0.104 | 7402 |

*Notes.* Risk aversion (B-EG) is a 6-point scale ranging from 1 = most risk-seeking (Lottery F) to 6 = most risk-averse (Lottery A) from the B–EG lottery task. Risk aversion (LA) is a 6-point scale ranging from 1 = most risk-seeking (Lottery F) to 6 = most risk-averse (Lottery A) from the loss-aversion version of the B–EG lottery task. Risk aversion (LA-binary) equals 1 if the respondent selected a lottery without a loss (Lotteries A–C) and 0 otherwise (Lotteries D–F), based on the loss-aversion version of the B–EG lottery task. Discount rate (1-month MPL) and discount rate (2-month MPL) are 7-point scales from the 1-month and 2-month multiple price list (MPL) tasks, respectively, where higher values indicate greater impatience. Planning horizon is a 7-point scale ranging from 0 (*"does not plan/plans day-to-day"*) to 6 (*"longer than 10 years"*), such that higher values indicate longer planning horizons. Planning horizon (binary) equals 1 if the respondent reports planning beyond the next year and 0 otherwise. $p < 0.05$ in bold.

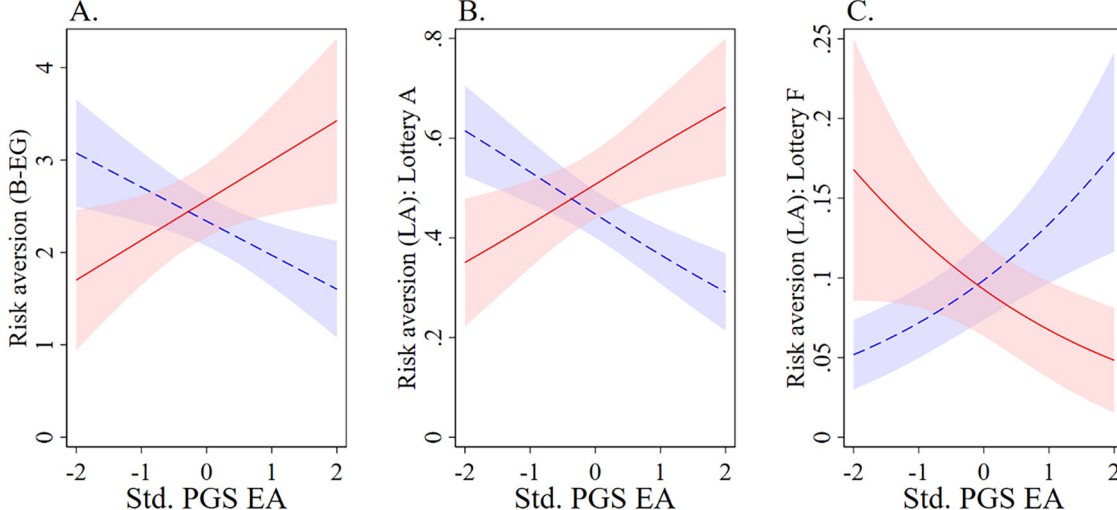

**Fig. 2 | Overview of risk-preference patterns by PGS EA and childhood disadvantage.** Panels (**A**–**C**) show predicted levels/probabilities of risk aversion across the range of standardized polygenic scores for educational attainment (PGS EA) in the experimental sample ($N = 624$). The red solid line represents respondents who experienced childhood disadvantage, and the blue dashed line represents respondents without childhood disadvantage; light red and light blue shaded regions depict 95% confidence intervals. Panel (**A**) plots predictions from the B–EG task based on participants' implied risk-aversion coefficients (Table 1). Panel **B** (**C**) shows predicted probability of choosing Lottery A (Lottery F) from the adapted B–EG lottery (Table 2), a 6-point scale ranging from 1 = most risk-seeking (Lottery F) to 6 = most risk-averse (Lottery A) that incorporates losses. All predictions control for age (in linear and quadratic form), sex, and the first 10 principal components of respondents' SNPs. Risk aversion is plotted on the vertical axis and standardized PGS EA on the horizontal axis.

[1.457, 2.870], meaning that the implied association of PGS EA and risk aversion for the 'disadvantage' group was positive, $\beta = 0.365$, SE = 0.149, $z = 2.44$, $p = 0.015$, OR = 1.440, 95% CI [1.075, 1.930]. For illustration, Panels B and C of Fig. 2 plot the predicted probability of choosing the most (least) risk-averse, Lottery A (F), across the standardized PGS EA distribution for both groups.

Collapsing the risk aversion (LA) 6-point scale into a dichotomous indicator equal to 1 for loss-free lotteries (Lottery A–C) and 0 for lotteries that include a loss (Lottery D–F) provides similar results. Analyzing risk aversion (LA-binary) by logistic regression, the main effect shows that for

respondents without childhood disadvantage, PGS EA significantly predicted lower risk aversion, $\beta = -0.543$, SE = 0.116, $z = -4.68$, $p < 0.001$, OR = 0.581, 95% CI [0.463, 0.730], with a significant positive interaction effect between PGS EA and childhood disadvantage, $\beta = 0.706$, SE = 0.205, $z = 3.45$, $p = 0.001$, OR = 2.027, 95% CI [1.356, 3.028], with the implied association of PGS EA and risk aversion for the 'disadvantage' group being positive but not significant, $\beta = 0.164$, SE = 0.169, $z = 0.97$, $p = 0.333$, OR = 1.178, 95% CI [0.846, 1.640]. The risk aversion (LA-binary) results remain robust when conditioning on choices in the standard B–EG risk task. The main effect again predicted a significant negative association of PGS EA

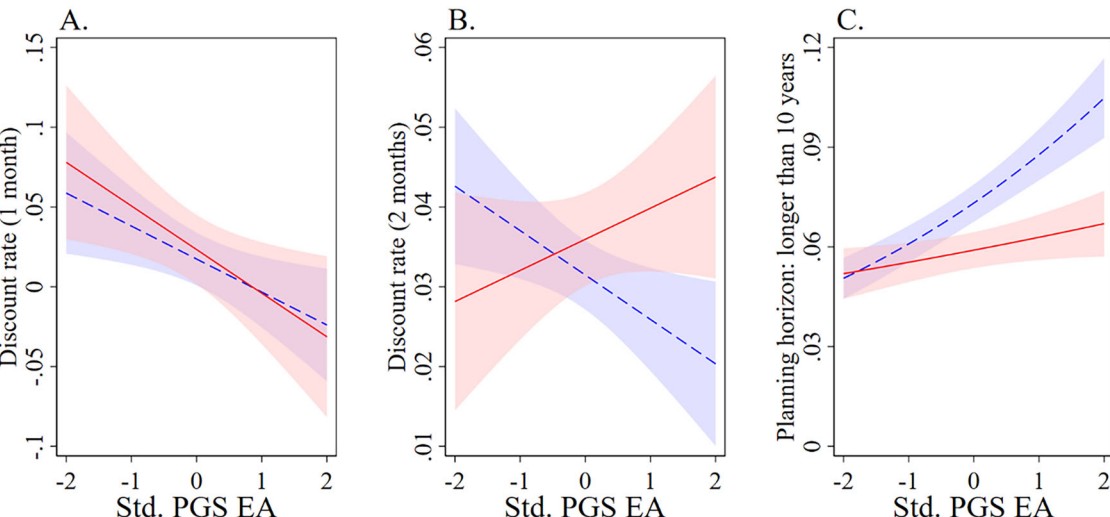

**Fig. 3 | Overview of time-preference patterns by PGS EA and childhood disadvantage.** Panels (**A**–**C**) show predicted levels/probabilities of time preferences across the range of standardized polygenic scores for educational attainment (PGS EA). In Panels (**A**) and (**B**), predictions are based on the experimental sample ($N = 624$); in Panel (**C**), predictions are based on the survey sample ($N = 5881$; 11,521 person-wave observations). The red solid line represents respondents who experienced childhood disadvantage, and the blue dashed line represents respondents without childhood disadvantage; light red and light blue shaded regions depict 95% confidence intervals. Panel (**A**) plots predicted discount rates from the 1-month

multiple price list (MPL) task, and Panel (**B**) plots predicted discount rates from the 2-month MPL task, based on participants' implied discount rates (Table 3). Panel (**C**) shows predicted probabilities of reporting a planning horizon longer than 10 years, from the 7-point planning horizon scale ranging from 0 ("*does not plan/plans day-to-day*") to 6 ("*longer than 10 years*"). All predictions control for age (in linear and quadratic form), sex, and the first 10 principal components of respondents' SNPs; predictions in Panel (**B**) also control for the respondents' discount rate from the 1-month MPL task. Standardized PGS EA is plotted on the horizontal axis in all panels.

and risk aversion, $\beta = -0.483$, SE $= 0.122$, $z = -3.96$, $p < 0.001$, OR $= 0.617$, 95% CI [0.485, 0.783], with a significant positive interaction effect, $\beta = 0.575$, SE $= 0.228$, $z = 2.52$, $p = 0.012$, OR $= 1.778$, 95% CI [1.137, 2.779], meaning that the implied association of PGS EA and risk aversion for the 'disadvantage' group was positive, $\beta = 0.092$, SE $= 0.193$, $z = 0.48$, $p = 0.633$, OR $= 1.096$, 95% CI [0.752, 1.599]. This indicates that the gene-by-environment interaction has an association with loss aversion, over and above the correlation with general risk tolerance.

Sensitivity power analysis indicates that, given the available sample size ($N = 624$), the risk-preference models had 80% power at $\alpha = 0.05$ to detect effects corresponding to a partial $R^2$ of approximately 0.013. This implies that, with the available data, effects explaining roughly 1.3% or more of the residual variance in the outcome could be detected with 80% probability.

Turning to time preferences: discount rates from the 1-month and 2-month multiple price list (MPL) tasks are analyzed using interval regression, based on the upper and lower bounds of participants' implied discount rates (Table 3), where higher values indicate greater impatience. For discount rates from the 1-month MPL, the main effect indicates that for respondents without childhood disadvantage, PGS EA significantly predicted lower discount rates, $z = -2.48$, $p = 0.013$, $\beta = -0.021$, SE $= 0.008$, 95% CI [−0.037, −0.004]. The interaction effect with childhood disadvantage was not significant, $z = -0.48$, $p = 0.633$, $\beta = -0.007$, SE $= 0.014$, 95% CI [−0.034, 0.021], meaning that for respondents with childhood disadvantage, PGS EA also significantly predicted lower discount rates, $z = -2.42$, $p = 0.016$, $\beta = -0.027$, SE $= 0.011$, 95% CI [−0.049, −0.005]. Panel A of Fig. 3 plots the relevant predicted discount rates, across the standardized PGS EA distribution for both groups. The results are similar for discount rates from the 2-month MPL. The main effect again predicted a significant negative genetic-related effect, $z = -3.69$, $p < 0.001$, $\beta = -0.011$, SE $= 0.003$, 95% CI [−0.018, −0.005], with a non-significant interaction effect, $z = 1.35$, $p = 0.177$, $\beta = 0.007$, SE $= 0.005$, 95% CI [−0.003, 0.017], meaning the implied effect for the 'disadvantage' group was negative, $z = -1.04$, $p = 0.299$, $\beta = -0.004$, SE $= 0.004$, 95% CI [−0.013, 0.004].

As an additional test, I estimated an interval regression on the discount rates from the 2-month MPL, controlling for the respondents' discount rates from the 1-month MPL (Panel B, Fig. 3). This estimation strategy allows us

to estimate the degree to which respondents are disproportionately impacted by longer delays. Here, the main effect suggests that a higher PGS EA statistically predicts a lower discount rate for respondents in the 'no disadvantage' group ($z = -2.41$, $p = 0.016$, $\beta = -0.006$, SE $= 0.002$, 95% CI [−0.010, −0.001]) whereas for the 'disadvantage' group the statistical effect was positive but not significant ($z = 1.29$, $p = 0.196$, $\beta = 0.004$, SE $= 0.003$, 95% CI [−0.002, 0.010]) with the difference being statistically significant ($z = 2.48$, $p = 0.013$, $\beta = 0.009$, SE $= 0.003$, 95% CI [0.002, 0.017]).

Further differences between the groups emerge when I turn to the survey sample ($N = 5881$; 11,521 person-wave observations), and the planning horizon measure. Analyzing the 7-point planning horizon scale —ranging from 0 ("*does not plan/plans day-to-day*") to 6 ("*longer than 10 years*"), such that higher values indicate longer planning horizons—using an ordered logistic regression, and cluster-robust standard errors, reveals that the childhood disadvantage significantly attenuates the relationship between PGS EA and the planning horizon. Specifically, the main effect suggests that a higher PGS EA is associated with a statistical increase in the planning horizon for the 'no disadvantage' group ($\beta = 0.199$, SE $= 0.025$, $z = 7.91$, $p < 0.001$, OR $= 1.220$, 95% CI [1.161, 1.282]) with a significant negative interaction effect between PGS EA and childhood disadvantage ($\beta = -0.131$, SE $= 0.040$, $z = -3.26$, $p = 0.001$, OR $= 0.877$, 95% CI [0.811, 0.949]), implying a smaller positive association of PGS EA and planning horizon for the 'disadvantage' group ($\beta = 0.068$, SE $= 0.031$, $z = 2.17$, $p = 0.030$, OR $= 1.070$, 95% CI [1.007, 1.138]). Panel C of Fig. 3 plots the predicted probability of reporting a planning horizon longer than 10 years, across the standardized PGS EA distribution for both groups.

Collapsing the 7-point scale into a dichotomous indicator—coded as 1 if the respondent reports planning beyond the next year and 0 otherwise—and estimating a logistic regression yields similar results. The main effect is positive and significant ($\beta = 0.196$, SE $= 0.027$ $z = 7.18$, $p < 0.001$, OR $= 1.216$, 95% CI [1.153, 1.283]) with a significant negative interaction effect ($\beta = -0.146$, SE $= 0.045$, $z = -3.25$, $p = 0.001$, OR $= 0.864$, 95% CI [0.791, 0.944]), implying a smaller positive association of PGS EA and planning horizon for the 'disadvantage' group ($\beta = 0.049$, SE $= 0.036$, $z = 1.37$, $p = 0.171$, OR $= 1.050$, 95% CI [0.979, 1.127]).

Sensitivity power analysis indicates that, in the experimental sample ($N = 624$), the time-preference models had 80% power at $\alpha = 0.05$ to detect effects corresponding to a partial $R^2$ of approximately 0.013. Given the larger sample size of the survey sample ($N = 11,521$), the models had 80% power at $\alpha = 0.05$ to detect very small effects, corresponding to a partial $R^2$ of approximately 0.001 or larger.

## Robustness

The main analyses face two potential concerns. First, between-family gene-by-environment designs may be biased if unobserved childhood environmental factors correlated with PGS EA—such as parental resources, time, or parenting style—also influence economic preferences[93–96]. To address this, I include the Parental Bonding Instrument, which measures respondents' retrospective experiences of parental care and overprotection before age 16[97,98]. The Parental Bonding Instrument focuses on parental care and overprotection, consisting of 14 items (7 for each parent) including "She/he let me do the things I liked doing"; "She/he made me feel I was not wanted" [reverse-coded]; and "She/he liked me to make my own decisions"; all rated on 1 ("strongly disagree") to 4 ("strongly agree") scales. I use the average score across items, where internal consistency reliability across the 14-items was good ($\alpha = 0.85$; average interitem correlation of 0.15). For non-responders (227 individuals in the experimental sample and 2025 individuals in the survey sample) I code missing values as zero and include a separate dichotomous control indicator for missingness (i.e., the missing indicator method). Although the relationship between PGS EA and the Parental Bonding Instrument is positive, the correlation is small and statistically non-significant for the experimental sample, $r(622) = 0.023$, $p = 0.568$, 95% CI [−0.056, 0.101], and small and statistically significant for the survey sample, $r(11,519) = 0.054$, $p < 0.001$, 95% CI [0.036, 0.072]. Importantly, the main results are robust to controlling for the Parental Bonding Instrument (Supplementary Tables 12 and 13).

A second concern is that PGS EA, while reflecting cognitive traits, also exhibits genetic variance overlap with personality domains such as Neuroticism and Openness[72], which have also been shown to be associated with risk-taking[99] and time preferences[100,101]. The core findings are robust to controlling for polygenic scores for the Big Five personality traits (Supplementary Tables 14 and 15; and see Supplementary Tables 16 and 17 for correlations between PGSs for personality traits and educational attainment), indicating that personality-related genetic overlap does not significantly account for the findings. To more directly isolate cognitive pathways, I also replace PGS EA with a polygenic score for IQ (PGS IQ) derived from Savage et al.[102], as well as respondents' phenotypic cognitive ability as measured in ELSA (see Supplementary Note 1). Despite the lower predictive accuracy of PGS IQ—owing to its smaller GWAS[103]—and the potential socioeconomic confounding of phenotypic cognitive ability, results using these alternative measures closely mirror the core findings (Supplementary Tables 18–21). This consistency strengthens confidence that my conclusions are not an artifact of the particular proxy used for cognitive-related traits.

Finally, following Dohmen et al.[9], I examine whether the estimated effects of PGS EA on economic preferences operate indirectly through adulthood socioeconomic channels, such as wealth, income, or education. Higher wealth and income may affect preferences by lowering the tangible and psychological costs of negative income shocks from risk-taking, or by reducing the need for immediate payoffs. Similarly, education may increase financial knowledge, which could help individuals better understand and evaluate probabilities and outcomes. Controlling for household wealth, income, and educational attainment does not significantly alter the main conclusions (Supplementary Tables 22 and 23).

## Discussion

Research has shown a robust link between cognitive ability and economic preferences[9–11] and how childhood disadvantage and poverty are associated with risk and time preferences[31–35,104,105]. I extend this literature by documenting how genetic variance associated with educational success interacts with childhood environments in the prediction of economic preferences, which are themselves powerful predictors of socioeconomic outcomes[1–8]. Understanding this interaction is important because individuals with similar cognitive-related genetic associations can be associated with very different decision-making preferences depending on their childhood environments. This speaks to broader questions about inequality and social mobility.

The results reveal an important yet disconcerting pattern. No matter what the level of PGS EA observed in this study, participants who experienced childhood disadvantage were more likely to have economic preferences in adulthood that correlate with social immobility[33]. Of course, as no economic decision involves only one preference or cognitive aspect, joint consideration of multiple factors is important[10,33]. For instance, individuals with lower PGS EA who experienced childhood disadvantage tend to be less risk-averse and less patient than their counterparts with similar genetic associations from more advantaged childhood environments. This combination of cognitive resources and economic preferences are robust predictors of addictive behaviors such as smoking and gambling, which combine intertemporal elements with risk considerations[106,107].

When childhood disadvantage is combined with high cognitive-related genetic associations the story is equally discouraging. Here, higher levels of cognitive-related resources are coupled with more risk-aversion and less patience than their counterparts with similar genetic associations from more advantaged childhood environments. Whilst it may be advantageous to couple lower cognitive resources with cautious tendencies in some domains[9], optimal financial investment decisions likely require a combination of a willingness to take calculated risks and high cognition[108,109]. Taken together, disadvantaged environments appear to limit the ability of individuals to capitalize on cognitive-related genetic associations, suggesting that environmental inequality can reinforce patterns of social immobility. This highlights the limitations of genetic determinism, illustrating how cognitive-related genetic associations interact dynamically with environmental factors, reinforcing the importance of nurturing environments for maximizing potential. This aligns with findings from childhood intervention programs, which illustrate that enriching early environments raises non-cognitive skills that promote success in social and economic life[110–112].

The results also provide an opening for perspectives on the mechanisms through which gene-by-environment interactions relate to economic preferences. In disadvantaged childhood environments, cognitive-related genetic resources may be directed toward detecting environmental threats, uncertainty, and resource scarcity—leading to greater risk aversion[63–65], as suggested by Adaptive Calibration Models[62]. In such contexts, risk-avoidant behavior may therefore reflect an adaptive response to environmental unpredictability and perceived threat[66,113–115]. Additionally, the finding that individuals from disadvantaged backgrounds exhibit similarly short-term (impatient) planning horizons across all PGS EA levels suggests that early adversity may impose pervasive constraints—such as scarcity-driven cognitive load or learned present-biased behaviors—that limit the expression of cognitive-related genetic associations in economic decision-making[116,117]. This pattern is consistent with the Experiential Canalization Framework[44,45], which posits that environmental adversity can interfere with the developmental processes and pathways that typically enable genetic predispositions to be associated with behavior—effectively muting trait differences in disadvantaged contexts. In contrast, in more advantaged childhood environments, the absence of such scarcity constraints appears to allow individuals with higher PGS EA to express economic preferences that are more aligned with findings from the cognitive ability literature[9]. Specifically, these individuals may be better able to override the impulsive, affect-driven tendencies—such as risk aversion and myopia—associated with System 1 processing[26–30].

These results may also help reconcile conflicting findings in the literature on early-life stress and economic preferences. Life history theory[36] predicts that early-life stress promotes "fast" strategies—impatience and risk-taking—while stable environments promote "slow" strategies characterized by patience and risk aversion. By contrast, uncertainty

management perspectives[41,42] argue that early-life stress predicts risk aversion and impatience because individuals adapt to anticipated future danger. Both perspectives have empirical support[40,42,43,118,119]. The findings help clarify these inconsistencies by identifying a boundary condition: childhood disadvantage aligns with the life-history prediction when paired with lower cognitive-related genetic associations but aligns with uncertainty management predictions when paired with higher cognitive-related genetic associations.

Finally, the findings speak to the longstanding endogeneity debate in the literature linking cognitive ability and economic preferences[9,10]. Economic preferences may affect investments in cognitive development, while cognitive ability may affect preferences by influencing the degree to which deliberative processes override impulsive responses[30]. Both may also be jointly determined by evolutionary pressures, which may have created tendencies for low cognitive skills to be coupled with cautious tendencies[9]. Although our analyses cannot establish causality, our results inform this debate by showing how cognitive-related genetic associations are related to economic preferences across diverse childhood environments.

## Limitations

Despite the robustness of these findings across multiple specifications, several limitations should be acknowledged when interpreting these results, particularly given that polygenic scores are best understood as predictive tools rather than causal explanations and are subject to well-known conceptual and methodological constraints[120,121]. First, the estimates remain based on comparisons between unrelated individuals. As with any between-family design, the estimated genetic associations with economic preferences may reflect residual confounding from unmeasured features of the family environment. Such dynastic effects arise when parental genotypes are associated with aspects of the childhood environment that are also related to offspring economic preferences. While I attempted to mitigate this concern by controlling for retrospective parental bonding, such measures may not fully capture the complexity of the early-life environment.

Future work using designs that leverage within-sibling variation in polygenic scores or that condition directly on parental genotypes would allow for a clearer separation of genetic and environmental contributions[76,122–124]. However, it should be noted that attenuation bias arising from measurement error in polygenic scores—which capture only a fraction of the underlying genetic architecture—tends to bias estimates toward zero, counteracting inflationary bias from unobserved environmental confounding[125]. Additionally, recall of childhood experiences may be imperfect, introducing measurement error that likely attenuates the estimated associations with early environments.

A second limitation concerns the interpretation of the educational attainment polygenic score (PGS EA). Although widely used, PGS EA captures genetic associations that extend beyond cognitive ability and overlaps with a range of non-cognitive traits, including aspects of personality. Importantly, my results remain stable when I control for polygenic scores for the Big Five personality traits and when I replicate the analyses using both a polygenic score for IQ and phenotypic cognitive ability. Nevertheless, it remains possible that other non-cognitive traits correlated with educational success—such as grit, motivation, curiosity, self-control, or perseverance[74,81,126–129]—contribute to the observed patterns. Separating cognitive from non-cognitive genetic associations remains a challenge and represents an important direction for future research.

A related limitation concerns the predictive strength of the polygenic scores available at the time of analysis. I rely on the educational attainment PGS from Lee et al.[70], which, although foundational, has since been substantially improved through larger and more fine-grained GWAS efforts (e.g., Okbay et al.[130]). The use of an older, less predictive score introduces attenuation bias: weaker genetic instruments reduce the power to detect gene–environment interactions and likely make these estimates more conservative. This limitation primarily concerns reliability rather than construct validity, as newer PGSs capture the same underlying educational attainment phenotype with greater precision. Accordingly, I do not expect the use of an earlier score to bias coefficient magnitudes in a directional way, apart from classical attenuation due to measurement error. This bias works against finding significant effects, suggesting that the reported estimates may, if anything, understate the true magnitude of the underlying relationships.

Finally, this study is limited to a high-income, industrialized context. Whether the same gene–environment patterns generalize to other societies —particularly those with different levels of environmental stability, economic development, and childhood adversity—remains an open question. Cross-national comparisons would be especially informative. Evidence from developing countries already suggests that the relationship between cognitive ability and economic preferences can differ across contexts: for example, Chowdhury et al.[68] find a negative association between IQ and patience, as well as a negative estimated effect of maternal education on risk tolerance. Their interpretation is that patience and risk taking are strategies that payoff in resource-rich, stable environments[2–4,7,8], whereas in resource-scarce or threatening environments, impatience and risk aversion may be more adaptive[41,42]. In both contexts, cognitive abilities are deployed to identify the strategies most likely to yield success.

## Data availability
The data that support the findings of this study are publicly available from the U.K. Data Archive: https://ukdataservice.ac.uk/. Download the following data files in Stata format: 5050 English Longitudinal Study of Ageing: Waves 0–10, 1998–2023; 8773 ELSA Polygenic Scores, 2022; 9081 Harmonised English Longitudinal Study of Ageing: ELSA-HCAP, 2018.

## Code availability
All Stata analysis scripts to replicate the results are available on OSF: osf.io/f8teh.

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

## Acknowledgements

I am grateful to Stephanie von Hinke for her very helpful comments. The author received no specific funding for this work.

## Competing interests

The author declares no competing interests.
