## [Transparent Peer Review file · Communications Psychology]

Associations between genetic variants for educational success and risk and time preferences vary by childhood environment

Corresponding Author: Professor Chris Dawson

Version 0:

Decision Letter:

Dear Professor Dawson,

Thank you for your patience during the peer-review process. Your manuscript titled "Economic Preferences, Genes, and Childhood Disadvantage" has now been seen by 2 reviewers, and I include their comments at the end of this message. They find your work of interest but raised some important points. We are interested in the possibility of publishing your study in *Communications Psychology*, but would like to consider your responses to these concerns and assess a revised manuscript before we make a final decision on publication.

We therefore invite you to revise and resubmit your manuscript, along with a point-by-point response to the reviewers. Please highlight all changes in the manuscript text file.

Editorially, we consider it important that the revised manuscript addresses the methodological and analytic concerns raised by the reviewers especially regarding moderation by educational attainment, selection of PGS for educational attainment, and standardization of variables. Please add a sensitivity power analysis, robustness check to account for the unstable estimates and predictive validity given the small sample. Please use a more recent PGS. R1 expresses strong concerns about the strength of evidence and we are also concerned about the sample size. We advise additional analysis and encourage you to explore if there is additional data available that could strengthen the work.

As you revise the manuscript in response to these issues, please also implement all requests in the attached Mandatory Revision Requests document. All requirements listed in this document need to be fully met, or the work will be returned to you for further revisions without peer review. This workflow is in place to increase the likelihood that the paper will be accepted for publication. It reduces the number of rounds of revision (and review) and ensures that the reviewers vet a version of the article that is compliant with journal policies. If you have any questions regarding the required revisions, please contact the journal prior to resubmission to avoid a negative outcome.

Please submit the following items:

- Revised manuscript
- Point-by-point response to the referees' comments
- Mandatory Revision Requests Table (attached).
- Cover letter (as a separate document)

via this link: Link Redacted .

** This url links to your confidential home page and associated information about manuscripts you may have submitted or are reviewing for us. If you wish to forward this email to co-authors, please delete the link to your homepage first **

Best regards,

Jennifer Bellingtier

Jennifer Bellingtier, PhD
Senior Editor
Communications Psychology

REVIEWER EXPERTISE:

Gene by environment interactions, polygenic scores, childhood disadvantage, risk taking

REVIEWER REPORTS:

Reviewer #1 (Remarks to the Author):

COMMSPSYCHOL-25-0734-T: Economic Preferences, Genes, and Childhood Disadvantage

This is a potentially important contribution to the gene-environment literature with respect to the mechanisms through which genetic polymorphisms may be differentially linked to a specific phenotype as a function of the environment in which one works, lives, or plays. The study design uses two data sources that demonstrate reliability and validity of the parameter estimates and provides new information regarding the role of risk aversion and time preference linking the moderation of the educational attainment PGS to educational attainment and cognitive ability, respectively. Most importantly, it provides tentative evidence for the fundamental nature of early life context in terms of how and for whom polygenic indices may be linked to their respective phenotypes.

That said, the evidence for environmental moderation (Table 4) of EA PGS effects on educational attainment or cognitive ability is only evident for educational attainment from the survey data using the ordered logit model for which evidence regarding the proportional odds assumption is not provided. Likewise, the GxE effects are only shown for the "University or college degree" category but the comparison group is quite different for each of the levels in this model especially if proportional odds is not met. Since they focus on the highest level of this variable, I would much rather see a logit model with "University or college degree" = 1 and all other levels below = 0. Then we get a simpler understanding of the estimated probabilities in the figure and the meaning of the parameter estimates.

The following analysis is very interesting and demonstrates that the EA PGS is a great predictor of the hypothesized mechanisms (time use/planning and risk aversion) and in opposite directions for those from different childhood backgrounds which is very much in line with existing theory and hasn't been shown clearly before.

However, as I mentioned already, the primary 'so what' is resting on some very thin evidence. Just one of the phenotypes in one of the studies. That is, for the other three models, there is no need to hypothesize what is behind the moderated genetic associations across the different environments, because there aren't any differences in the genetic associations across environments.

Finally, for the one model that they do show the main GxE (Table 4, Model 3), they never return to this model to show how the two factors introduced in subsequent modeling help explain the GxE parameter estimate. Some call this mediated moderation -- what is it about the disadvantaged vs. non disadvantaged environments that explains (mediates) their differential association (moderation) of genotype and phenotype? I simply want to see Table 4 Model 3 estimate ($b = -.108$, $p < .001$) with controls for time use/planning and risk aversion. Whether or not these controls are included as independent or as an interaction term is up to the authors (I'd personally like to see it both ways). Then show me how much $-.108$ decreases in magnitude (assuming the two covariates are mean centered if allowed as interactions). Thus, these would provide the evidence they seek to provide. Otherwise, it's still interesting results but they have nothing to do with the GxE association shown in Table 4, Model 3. I hope that makes sense.

Reviewer #2 (Remarks to the Author):

The paper investigates how genetic and environmental factors jointly shape economic preferences, using data from ELSA and an additional survey sample. The major claim is that the effects of educational polygenic scores (PGS) on risk and time preferences are moderated by family circumstances in childhood. The study combines behavioral experimental data with genomic information and socioeconomic background. The topic is timely and relevant for researchers in behavioral genetics, developmental psychology, and behavioral economics.

The central idea of linking gene–environment interplay to economic decision-making is interesting, and the paper makes a useful contribution to integrating these literatures. The evidence is broadly consistent with expectations but appears in parts statistically weak. The results on cognitive ability are not fully integrated into the argument and need further clarification.

The analyses represent standard Gene \times Environment modeling in the social sciences. The study relies on the Lee et al. (2018) GWAS-based educational attainment PGS, which was state-of-the-art at the time but has since been improved (e.g., Okbay et al., 2022). While this does not undermine the results, it limits the predictive strength of the analyses. If possible, it would be good to use a newer PGS; if not, the authors should discuss how a more predictive score might affect the findings. Reporting z-standardized coefficients and clarifying the PGS scaling would improve interpretability. The description of methods is detailed, but some technical information on PGS construction and robustness checks should be moved to the supplement.

It is good that the authors acknowledge the possibility of gene–environment correlation (rGE). The analysis could, however, be strengthened by showing, for example, correlations between the educational attainment PGS and childhood disadvantage, which would provide an indication of the extent of rGE. The authors could also discuss whether age might pose a problem. Mortality effects seem minor, but recall bias could be an issue.

The manuscript would benefit from being more concise and focused. Some sections could be streamlined, and several methodological clarifications would strengthen the interpretation of the results. The extensive theoretical background makes it hard to keep the focus on the main research questions. The frequent use of direct quotations and footnotes further interrupts the reading flow. Integrating references more smoothly or reducing their number would improve readability. The introduction, for example, could be shortened by focusing on the research gap, key hypotheses, and main contribution. Opening the paper with a quotation from Francis Galton is rather unusual and may not work well for an empirical article. In addition, including an overview table somewhere in the paper summarizing the different analyses, their samples, and analytical aims would help readers to follow the study's overall logic and structure.

Given the manuscript's length, several technical sections could be moved to the appendix. Details on the construction of the polygenic score, the validation analyses on cognitive ability, and robustness checks could be presented as supplementary materials.

The description of the PGS could be improved, as it is now well established that the educational attainment PGS reflects both cognitive and non-cognitive traits (e.g., Demange et al., 2021). In addition, the use of the term “genetic potential” is misleading; it would be better to refer to the effects of genetic variants or genetic influences on education.

The descriptive tables in the appendix suggest that the polygenic score for educational attainment was not standardized prior to analysis (values range roughly between 4,190 and 4,300). For comparability and interpretability, it would be preferable to z-standardize the PGS. In addition, the reported age range in the survey-based sample (31–90 years) appears inconsistent with ELSA's 50+ sampling frame and should be clarified or corrected.

The role of Table 4 in the overall argument could be clearer. It is not entirely obvious whether the analyses on cognitive ability are meant as a conceptual part of the main hypotheses or merely as a validation step for the educational PGS. The reported coefficients appear not to be z-standardized. For comparison across models, it would be useful to include standardized betas.

In general, applying an EA PGS to this sample is methodologically appropriate. However, the experimental sample with PGS data ($N = 624$) is at the lower end of what is typically adequate for polygenic score analyses, which may result in unstable estimates and wide confidence intervals. This is particularly concerning as the authors test gene-environment interactions, which typically require substantially larger samples to achieve adequate statistical power. Moreover, the predictive validity of the PGS may be reduced in this sample due to both the small sample size and the older age of participants (mean age 64). This is something the authors should explicitly discuss. They should also acknowledge that the limited sample size reduces statistical power and address these factors as potential constraints on the predictive accuracy of the PGS.

Some degree of internal validation might be possible using the larger survey-based sample. In addition, the authors should explicitly acknowledge the limitations of their design and present their results as preliminary, calling for replication in independent samples.

Additionally, it is possible that the dataset only provides PGS based on older GWAS results (rather than the most recent one, Okbay et al., 2022). If it is not possible to use a more recent PGS, the authors should explicitly discuss the potential implications for their findings (e.g., lower prediction compared to more recent scores).

as a supplementary peer review file. However, on author request, confidential information and data can be removed from the published reviewer reports and rebuttal letters prior to publication. If your manuscript has been previously reviewed at another journal, those Reviewers' comments would not form part of the published peer review file.

Version 1:

Decision Letter:

** Please ensure you delete the link to your author homepage in this e-mail if you wish to forward it to others **

Dear Professor Dawson,

Your manuscript titled "Childhood environments alter how genetic propensities shape adult economic preferences" has now been seen by our reviewers, whose comments appear below. In light of their advice I am delighted to say that we are happy, in principle, to publish a suitably revised version in Communications Psychology.

We therefore invite you to revise your paper one last time to address the remaining concerns of our reviewers and a list of editorial requests. At the same time we ask that you edit your manuscript to comply with our format requirements and to maximise the accessibility and therefore the impact of your work.

EDITORIAL REQUESTS:

Please review our specific editorial comments and requests regarding your manuscript in the attached edited manuscript and "Editorial Requests Table". Please outline your response to each request in the right hand column. Please upload the completed table with your manuscript files as a Related Manuscript file.

SUBMISSION INFORMATION:

OPEN ACCESS:

* **CODE AVAILABILITY:** All Communications Psychology manuscripts must include a section titled "Code Availability" at the

end of the methods section. We require that the custom analysis code supporting your conclusions is made available in a publicly accessible repository at this stage; please choose a repository that generates a digital object identifier (DOI) for the code; the link to the repository and the DOI must be included in the Code Availability statement. Publication as Supplementary Information will not suffice.

*** DATA AVAILABILITY:**

Link Redacted

Best regards,

Jennifer Bellingtier

Jennifer Bellingtier, PhD
Senior Editor
Communications Psychology

REVIEWER EXPERTISE:

Gene by environment interactions, polygenic scores, childhood disadvantage, risk taking

REVIEWERS' COMMENTS:

Reviewer #1 (Remarks to the Author):

Communications Psychology: COMMSPSYCHOL-25-0734A "Childhood environments alter how genetic propensities shape adult economic preferences"

I am very grateful to the author(s) for providing such a more clear draft in the revised version of the manuscript. My original enthusiasm for the 1st submission started to change the more that I read and the more I tried to tell a coherent story in light of the empirical evidence that was provided. However, my enthusiasm remained throughout my reading of the revised paper and the presentation of the results and ancillary analyses. It tells a clear, consistent, honest, and important story and highlights the important contributions that were made on this paper that were not told as clearly in the original submission. They are equally clear about the limitations based on the current structure and they provide very doable next steps for the scientific community.

I only have a small number of remaining issues that make me scratch my head. I don't see any as required. Indeed, they address each but I feel like a small amount needs to be done to make the typical reader satisfied.

1. They are quite clear that there are some limitations due to low power. I appreciate the efforts to point this out. But some small statements might help us to better understand what they mean when they say "power." Power to detect an interaction of what size? Is this beta (the size of the interaction term) for the interaction is in line with the theory provided? There are several different GxE models that predict different types of main and interactive effects? In most cases we only care about the GxE term but "telling a story" involves the main effect (at a value of zero), the genetic effect, and the GxE term. Then show me the power. Many are particularly concerned about these producing significant effects as evidence of type-II errors. The cross-over results are very important, but that specific test may not accurately test the power being evaluated which requires the value AND direction of the main effect and the interaction term. To highlight, based on these results, there would be no main genetic effect without the interaction ($b=0$) with the cross-over model which is a very strong version of the differential susceptibility theory. Again, just something to think about when summarizing the results. I still find them fascinating, just consider how you might convince people what your power is actually testing and how it helps you to differentiate from complete noise.

2. They have a nice statement about the new risk score, and they only mention reliability and no discussion of validity. Is there any reason why there would be bias in the magnitude of the coefficient because of what it is measuring that the other wasn't. Again, not necessary, just something to think about.

3. I would like to know why knowing the context is so important for these results matters. I very much agree with the statement in the conclusion, but I'd like to know why. This is very important but there are a lot of things about this statement

that could be interpreted differently and HOW would individual identities give us clues about the social moderation in that larger context as to the WHYs and the WHOs. Again, not a necessary statement, but it will resonate with a much larger readership when you give us more information about the 'so what.' I like what is included I simply wonder if a few more sentences can be added to truly show the significance of this type of work to scientific audience, writ large.

Reviewer #2 (Remarks to the Author):

My original concerns have been addressed. In particular, the clarifications regarding the construction and interpretation of the PGS, the discussion of rGE and design limitations, and the more cautious framing of the power issues have substantially improved the manuscript. I do not have further major methodological concerns. I have only two comments left.

First, I noticed that the title has been changed. While the revised version is more specific, I find that it overstates the strength of the evidence. The term "alter" implies a causal claim that goes beyond what the data can support. As the authors themselves acknowledge, several features of the study limit causal inference, including the between-family design and the modest statistical power of some key interaction terms. I would therefore suggest revising the title perhaps closer in tone to the original version.

Second, although I did not raise this point in my first review, I recommend adding two to three references in the discussion that provide a general overview of polygenic score limitations (e.g., environmental confounding, population stratification, GWAS sample biases, assortative mating). While the authors already discuss several study-specific limitations, a brief general reference would help orient readers who are less familiar with the genetics literature.

Thank you for your comments. We really appreciate them. Your report has prompted us to reexamine and clarify our views and has generated fresh insights leading to a thoroughly revised paper. Our responses to your specific comments are below.

This is a potentially important contribution to the gene-environment literature with respect to the mechanisms through which genetic polymorphisms may be differentially linked to a specific phenotype as a function of the environment in which one works, lives, or plays. The study design uses two data sources that demonstrate reliability and validity of the parameter estimates and provides new information regarding the role of risk aversion and time preference linking the moderation of the educational attainment PGS to educational attainment and cognitive ability, respectively. Most importantly, it provides tentative evidence for the fundamental nature of early life context in terms of how and for whom polygenic indices may be linked to their respective phenotypes.

That said, the evidence for environmental moderation (Table 4) of EA PGS effects on educational attainment or cognitive ability is only evident for educational attainment from the survey data using the ordered logit model for which evidence regarding the proportional odds assumption is not provided. Likewise, the GxE effects are only shown for the “University or college degree” category but the comparison group is quite different for each of the levels in this model especially if proportional odds is not met. Since they focus on the highest level of this variable, I would much rather see a logit model with “University or college degree” = 1 and all other levels below = 0. Then we get a simpler understanding of the estimated probabilities in the figure and the meaning of the parameter estimates.

Table 4 is really there as a test of validity of the polygenic score (PGS) for educational attainment (EA), illustrating how it predicts the associated phenotypes. The way it was written in the first submission certainly does not make this as clear as it should be. Indeed, the Scarr-Rowe narrative is somewhat peripheral to the paper’s primary contribution, which is to demonstrate that early-life environments do not merely constrain genetic expression, but channel cognitive genetic resources toward behavioral strategies (i.e., economic preferences) that are adaptive within those contexts. We now relegate the original Table 4 to Supplemental Material B and discuss the analysis briefly, in the context of PGS validity, in the “Polygenic Score (PGS) for Educational Attainment (EA)” section of the paper. We also remove the long discussion of Scarr-Rowe in introduction. The introduction is now more focused on our central contribution. Supplemental Material B also includes your suggestion of Scarr-Rowe analysis using a logit model with “University or college degree” = 1 and all other levels below = 0. Indeed, using this approach we no longer find any evidence of moderation.

The following analysis is very interesting and demonstrates that the EA PGS is a great predictor of the hypothesized mechanisms (time use/planning and risk aversion) and in opposite directions for those from different childhood backgrounds which is very much in line with existing theory and hasn’t been shown clearly before.

Thank you, and yes, this is the central result of the paper. As also noted by Reviewer 2, Table 4 and the related analysis is somewhat of a distraction from the main aims of the paper. As stated above, the Scarr–Rowe narrative is somewhat peripheral to the paper’s primary contribution, which is to demonstrate that early-life environments do not merely constrain genetic expression, but channel cognitive genetic resources toward behavioral strategies that are adaptive within those contexts.

However, as I mentioned already, the primary ‘so what’ is resting on some very thin evidence. Just one of the phenotypes in one of the studies. That is, for the other three models, there is no need to hypothesize what is behind the moderated genetic associations across the different environments, because there aren’t any differences in the genetic associations across environments.

As above, the Scarr-Rowe narrative is somewhat peripheral to the paper’s primary contribution. However, the “very thin evidence” (i.e., the absence of Scarr–Rowe effects in our data) supports the main interpretation of our core results.

Specifically, if the Scarr–Rowe pattern were present—meaning that the genetic potential for cognitive ability is more fully expressed in enriched, advantaged environments—our core findings could be explained by a simpler account: children from disadvantaged backgrounds may have fewer opportunities to develop cognitive skills, and because cognitive ability is associated with greater patience and risk tolerance, the moderated genetic associations we observe could simply reflect differences in cognitive development.

However, our analyses do not show Scarr–Rowe moderation of the educational attainment PGS. This absence suggests that the differential genetic associations we observe are not reducible to advantaged children realizing more of their cognitive potential. Instead, the results align more closely with theoretical frameworks emphasizing directional channelling rather than blocked expression.

First, the Experiential Canalization Framework proposes that childhood adversity heightens emotional reactivity, chronic stress, negative affect, and reduces regulatory control—all of which promote reliance on fast, intuitive (System 1) decision-making typically associated with risk aversion and impatience, at the expense of deliberative (System 2) processes associated with patience and risk neutrality.

Second, Adaptive Calibration Models argue that early environments calibrate biological and behavioral systems in ways that are developmentally adaptive. Under this framework, genetic propensities are not suppressed but channelled differently depending on early-life cues: in advantaged settings, genetic cognitive resources may be expressed through long-term planning and opportunity-seeking, whereas in disadvantaged settings, the same genetic resources may be directed toward vigilance, threat detection, and short-term security.

Taken together, the lack of Scarr–Rowe effects support the interpretation that early environments shape how genetic cognitive resources are deployed, rather than whether they can be expressed. This strengthens the theoretical contribution of the paper: the moderated genetic associations we observe are consistent with experiential canalization and adaptive

calibration, not simply with differential cognitive-skill development across environments. As mentioned above the Scarr-Rowe analysis is relegated to Supplemental Material B.

Finally, for the one model that they do show the main GxE (Table 4, Model 3), they never return to this model to show how the two factors introduced in subsequent modeling help explain the GxE parameter estimate. Some call this mediated moderation -- what is it about the disadvantaged vs. non disadvantaged environments that explains (mediates) their differential association (moderation) of genotype and phenotype? I simply want to see Table 4 Model 3 estimate ($b = -.108, p < .001$) with controls for time use/planning and risk aversion. Whether or not these controls are included as independent or as an interaction term is up to the authors (I'd personally like to see it both ways). Then show me how much $-.108$ decreases in magnitude (assuming the two covariates are mean centered if allowed as interactions). Thus, these would provide the evidence they seek to provide. Otherwise, it's still interesting results but they have nothing to do with the GxE association shown in Table 4, Model 3. I hope that makes sense.

As stated above, we agree that Table 4 is distracting in its current form. Reviewer 2 also noted that *“the role of Table 4 in the overall argument could be clearer. It is not entirely obvious whether the analyses on cognitive ability are meant as a conceptual part of the main hypotheses or merely as a validation step for the educational PGS.”*

In our manuscript, the purpose of the Table 4 Scarr-Rowe models was solely to validate the educational attainment PGS by showing whether it correlates with the cognitive phenotypes. These models were not intended as part of our core theoretical argument. Our main focus is on how early-life environments interact with genetic propensities to channel cognitive resources toward or away from patience and risk-taking.

We agree that presenting the Scarr-Rowe models in the main text risks confusing readers about their substantive role. To improve clarity, we have moved these analyses to Supplemental Material B and now report only the relevant correlations in the PGS section as a validity check.

The broader question you raise is interesting but falls outside the scope of the present study. Addressing it would require strong assumptions about exogeneity, since both phenotypic educational attainment and phenotypic cognitive skills may themselves influence time and risk preferences. This is consistent with the broader literature on feedback loops between human capital formation and economic preferences (e.g., Heckman & Mosso, 2014).

Heckman, J. J., & Mosso, S. (2014). The economics of human development and social mobility. *Annu. Rev. Econ.*, 6(1), 689-733.

Thank you for your comments. We really appreciate them. Your report has prompted us to reexamine and clarify our views and has generated fresh insights leading to a thoroughly revised paper. Our responses to your specific comments are below.

The paper investigates how genetic and environmental factors jointly shape economic preferences, using data from ELSA and an additional survey sample. The major claim is that the effects of educational polygenic scores (PGS) on risk and time preferences are moderated by family circumstances in childhood. The study combines behavioral experimental data with genomic information and socioeconomic background. The topic is timely and relevant for researchers in behavioral genetics, developmental psychology, and behavioral economics.

The central idea of linking gene–environment interplay to economic decision-making is interesting, and the paper makes a useful contribution to integrating these literatures. The evidence is broadly consistent with expectations but appears in parts statistically weak. The results on cognitive ability are not fully integrated into the argument and need further clarification.

We agree that the phenotypic cognitive ability variable is not well integrated into the paper. We discuss this in more detail below in response to your further comments. In short, the use of phenotypic cognitive ability was to validate the polygenic score for educational attainment, and later as a robustness check for our main results. In this view, we have now substantially revised the manuscript.

In response to “in parts statistically weak” we now include post-hoc power analysis, which indicates that our main results are indeed generally well powered ($\approx 80\text{--}99\%$) at $\alpha = 0.05$.

The analyses represent standard Gene \times Environment modeling in the social sciences. The study relies on the Lee et al. (2018) GWAS-based educational attainment PGS, which was state-of-the-art at the time but has since been improved (e.g., Okbay et al., 2022). While this does not undermine the results, it limits the predictive strength of the analyses. If possible, it would be good to use a newer PGS; if not, the authors should discuss how a more predictive score might affect the findings. Reporting z-standardized coefficients and clarifying the PGS scaling would improve interpretability. The description of methods is detailed, but some technical information on PGS construction and robustness checks should be moved to the supplement.

We now acknowledge in the ‘Limitations’ section that the Lee et al. (2018) has been improved upon but that it is the latest available version available ELSA researchers. We also now discuss the implications. In short, although newer GWAS (e.g., Okbay et al., 2022) yield more predictive scores, using a less predictive PGS does not bias $G \times E$ estimates; rather, it attenuates effect sizes toward zero due to measurement error. Thus, our results may be considered conservative. A sensitivity-based power analysis indicated that the study retained 80–99% power across most of the models even with the older PGS. We therefore expect that using a more predictive PGS would increase precision but would not alter the substantive conclusions.

It should be noted that we do standardize the polygenic score—and thus, report z-standardized coefficients—however when we report the descriptive statistics the “unadjusted” scores are summarized. Summarizing the standardized scores would lead to a table of zeros (means) and ones (standard deviations), which would not be informative. We do though make it clearer in the manuscript that the polygenic scores and other important predictors are standardized, to facilitate interpretation of the results. We agree, we were not clear about standardization in our first submission.

It is good that the authors acknowledge the possibility of gene–environment correlation (rGE). The analysis could, however, be strengthened by showing, for example, correlations between the educational attainment PGS and childhood disadvantage, which would provide an indication of the extent of rGE. The authors could also discuss whether age might pose a problem. Mortality effects seem minor, but recall bias could be an issue. The manuscript would benefit from being more concise and focused. Some sections could be streamlined, and several methodological clarifications would strengthen the interpretation of the results. The extensive theoretical background makes it hard to keep the focus on the main research questions. The frequent use of direct quotations and footnotes further interrupts the reading flow. Integrating references more smoothly or reducing their number would improve readability. The introduction, for example, could be shortened by focusing on the research gap, key hypotheses, and main contribution. Opening the paper with a quotation from Francis Galton is rather unusual and may not work well for an empirical article. In addition, including an overview table somewhere in the paper summarizing the different analyses, their samples, and analytical aims would help readers to follow the study’s overall logic and structure.

We appreciate this helpful comment and have strengthened the manuscript accordingly.

First, and in response to “gene–environment correlation (rGE)”, we did in the first submission present evidence of the relationship between the educational attainment PGS and childhood environment. Specifically, Figure 4 in our first submission displayed the distribution of EA-PGS scores separately for individuals from advantaged and disadvantaged backgrounds. In both samples, individuals from disadvantaged backgrounds have EA-PGS scores approximately 0.097–0.101 SD lower than the overall sample mean. This pattern is expected, given that individuals inherit their genetic makeup from parents whose own genetic traits contribute to their educational attainment, socioeconomic resources, and the environments they provide.

In addition, we now report the correlation between the EA-PGS and childhood disadvantage in the PGS section. This provides a direct and transparent indication of the magnitude of rGE in our data—which is importantly, very small.

Second, we now explicitly acknowledge recall bias as a limitation. Childhood disadvantage is measured retrospectively, and misremembering would introduce measurement error. As we note in the revised discussion, such error would generally attenuate associations, making it harder—not easier—to detect the $G \times E$ patterns we observe.

Third, we have streamlined the manuscript substantially. The Scarr–Rowe analyses—which were included only as a validity check and were not central to our theoretical argument—have been moved to Supplemental Material B. This allows us to maintain focus on the paper’s main

contribution: how early-life environments interact with genetic propensities to channel cognitive resources toward or away from patience and risk-taking.

Fourth, we have reduced quotations and footnotes, removed the opening Galton quotation, and tightened the theoretical background to keep the focus on the core research questions. These revisions improve readability and reduce distraction from the main argument.

Finally, given the new streamlined structure, an additional overview table is no longer necessary—the sequence of analyses is now shorter and more clearly motivated within the text. Should the editor prefer, we would be happy to add such a table, but in the current format it does not add clarifying value.

Given the manuscript's length, several technical sections could be moved to the appendix. Details on the construction of the polygenic score, the validation analyses on cognitive ability, and robustness checks could be presented as supplementary materials. The description of the PGS could be improved, as it is now well established that the educational attainment PGS reflects both cognitive and non-cognitive traits (e.g., Demange et al., 2021). In addition, the use of the term “genetic potential” is misleading; it would be better to refer to the effects of genetic variants or genetic influences on education.

In line with our broader efforts to streamline the manuscript, we have moved the validation analyses (previously Table 4 / Scarr–Rowe analyses), and robustness checks—to the Supplemental Material A and B. This keeps the main text focused on our core theoretical contribution while still providing full methodological transparency.

We have also improved the description of the educational attainment PGS. Specifically, we now note that it reflects both cognitive and non-cognitive traits (Demange et al., 2021). Finally, following your suggestion, we have replaced the term “genetic potential” with “genetic influences on educational attainment” or “cognitive-related genetic propensities” throughout the manuscript to better reflect the nature of polygenic effects.

The descriptive tables in the appendix suggest that the polygenic score for educational attainment was not standardized prior to analysis (values range roughly between 4,190 and 4,300). For comparability and interpretability, it would be preferable to z-standardize the PGS. In addition, the reported age range in the survey-based sample (31–90 years) appears inconsistent with ELSA's 50+ sampling frame and should be clarified or corrected. The role of Table 4 in the overall argument could be clearer. It is not entirely obvious whether the analyses on cognitive ability are meant as a conceptual part of the main hypotheses or merely as a validation step for the educational PGS. The reported coefficients appear not to be z-standardized. For comparison across models, it would be useful to include standardized betas.

We agree that Table 4 in the original submission was distracting, and indeed Reviewer 1 also found it confusing. As noted, Table 4 was included solely to validate the educational attainment PGS, not as part of the core hypotheses. To reduce confusion and better focus the manuscript, we have moved these analyses to Supplemental Material B, reporting only the relevant correlations in the PGS section as a validity check.

Regarding standardization, summarizing z-standardized PGS scores in a table would produce a table of means of approximately zero and standard deviations of one, which is not particularly informative. We do, however, now make it explicit in the manuscript that the PGS and other key predictors are standardized, to facilitate interpretation and comparability across models.

Finally, regarding the age range in the survey-based sample: although ELSA's core sampling frame is 50+, the dataset does include some participants under 50—specifically cohabiting partners and spouses of core members, as well as a younger partner cohort included in certain early waves and later reclassified. We have clarified this in the manuscript.

In general, applying an EA PGS to this sample is methodologically appropriate. However, the experimental sample with PGS data ($N = 624$) is at the lower end of what is typically adequate for polygenic score analyses, which may result in unstable estimates and wide confidence intervals. This is particularly concerning as the authors test gene-environment interactions, which typically require substantially larger samples to achieve adequate statistical power. Moreover, the predictive validity of the PGS may be reduced in this sample due to both the small sample size and the older age of participants (mean age 64). This is something the authors should explicitly discuss. They should also acknowledge that the limited sample size reduces statistical power and address these factors as potential constraints on the predictive accuracy of the PGS.

We thank the reviewer for this important point. We agree that the experimental sample ($N = 624$) is relatively small for PGS analyses and that testing gene-environment interactions in smaller samples can reduce power and yield wider confidence intervals. We explicitly acknowledge this limitation in the revised manuscript. Importantly, the limited predictive validity of the PGS in this context makes detecting significant effects harder, so our findings can be viewed as conservative estimates rather than inflated effects.

To provide transparency regarding power, we now include post-hoc sensitivity power analyses. In short, the main effects and interactions were generally well powered ($\approx 80\text{--}99\%$) at $\alpha = 0.05$.

These analyses underscore that, despite the modest sample size, our main results are robust, and any non-significant interactions are likely conservative due to reduced power rather than methodological artifacts.

Some degree of internal validation might be possible using the larger survey-based sample. In addition, the authors should explicitly acknowledge the limitations of their design and present their results as preliminary, calling for replication in independent samples.

We originally included internal validation in both the experimental and survey-based samples. Specifically, we show that the educational attainment (EA) PGS is positively correlated with both phenotypic educational attainment and cognitive ability in both samples, confirming that the PGS behaves as expected. In addition, the distribution PGS EA is similar across childhood advantage/disadvantage samples, supporting the generalizability of our findings. These analyses provide confidence that the PGS captures meaningful variation in cognitive and educational traits and that our main results are not driven by anomalies in the smaller experimental sample.

We also explicitly acknowledge the limitations of our design. The relatively modest size of the experimental sample ($N = 624$) limits power, particularly for some interaction terms, and the retrospective measurement of childhood disadvantage may introduce recall bias, which would generally attenuate associations. Taken together, these factors make our estimates conservative, and we present our findings as preliminary, emphasizing the importance of replication in independent samples.

Additionally, it is possible that the dataset only provides PGS based on older GWAS results (rather than the most recent one, Okbay et al., 2022). If it is not possible to use a more recent PGS, the authors should explicitly discuss the potential implications for their findings (e.g., lower prediction compared to more recent scores).

We thank the reviewer for this comment. As noted in our previous responses, we used the Lee et al. (2018) educational attainment PGS, which was state-of-the-art at the time of data collection. While more recent GWAS (e.g., Okbay et al., 2022) yield more predictive scores, using an older PGS does not bias our results, but rather reduces statistical power and attenuates effect sizes toward zero. Thus, our findings can be considered conservative. We now explicitly discuss this in the manuscript, noting that a more predictive score would likely increase precision but would not change the substantive interpretation of the results.

Thank you for your comments. My responses to your specific comments are below.

I am very grateful to the author(s) for providing such a more clear draft in the revised version of the manuscript. My original enthusiasm for the 1st submission started to change the more that I read and the more I tried to tell a coherent story in light of the empirical evidence that was provided. However, my enthusiasm remained throughout my reading of the revised paper and the presentation of the results and ancillary analyses. It tells a clear, consistent, honest, and important story and highlights the important contributions that were made on this paper that were not told as clearly in the original submission. They are equally clear about the limitations based on the current structure and they provide very doable next steps for the scientific community.

Thank you. Your original comments were very helpful in clarifying the papers main contribution.

I only have a small number of remaining issues that make me scratch my head. I don't see any as required. Indeed, they address each but I feel like a small amount needs to be done to make the typical reader satisfied.

1. They are quite clear that there are some limitations due to low power. I appreciate the efforts to point this out. But some small statements might help us to better understand what they mean when they say "power." Power to detect an interaction of what size? Is this beta (the size of the interaction term) for the interaction is in line with the theory provided? There are several different GxE models that predict different types of main and interactive effects? In most cases we only care about the GxE term but "telling a story" involves the main effect (at a value of zero), the genetic effect, and the GxE term. Then show me the power. Many are particularly concerned about these producing significant effects as evidence of type-II errors. The cross-over results are very important, but that specific test may not accurately test the power being evaluated which requires the value AND direction of the main effect and the interaction term. To highlight, based on this these results, there would be no main genetic effect without the interaction ($b=0$) with the cross-over model which is a very a strong version of the differential susceptibility theory. Again, just something to think about when summarizing the results. I still find them fascinating, just consider how you might convince people what your power is actually testing and how it helps you to differentiate from complete noise.

In light of your comments and the editor's guidance ("Please do not report post-hoc power based on the effect observed in your study. Instead, please calculate a sensitivity power analysis to determine the smallest effect your study was powered to detect given your sample"), we have changed:

"Post-hoc power calculations based on the observed effect sizes (partial R^2) indicate that the main effects and interactions in the risk-preference models were generally well powered (approximately 80–99%) at $\alpha = 0.05$, with the exception of the interaction in the final model, which had moderately lower sensitivity ($\approx 73\%$), indicating reduced power to detect effects of that magnitude."

To:

“Sensitivity power analysis indicates that, given the available sample size ($N = 624$), the risk-preference models had 80% power at $\alpha = 0.05$ to detect effects corresponding to a partial R^2 of approximately 0.013. This implies that, with the available data, effects explaining roughly 1.3% or more of the residual variance in the outcome could be detected with 80% probability.”

And changed for the time preference results, from:

“Post-hoc power calculations based on the observed effect sizes (partial R^2) indicate that the main effects and the interactions in the time preference models were generally well powered (approximately 80–99%) at $\alpha = 0.05$. However, the interaction terms in the experimental discount rate models were low-powered, and their non-significant coefficients should therefore be interpreted cautiously, as limited power reduces the ability to distinguish small or moderate effects from zero. The exception is when estimating the discount rate for a 2-month delay conditional on the discount rate for a 1-month delay: here the main effect and interaction term both had moderate power ($\approx 70\%$).”

To:

“Sensitivity power analysis indicates that, in the experimental sample ($N = 624$), the time-preference models had 80% power at $\alpha = 0.05$ to detect effects corresponding to a partial R^2 of approximately 0.013. Given the larger sample size of the survey sample ($N = 11,521$), the models had 80% power at $\alpha = 0.05$ to detect very small effects, corresponding to a partial R^2 of approximately 0.001 or larger.”

2. They have a nice statement about the new risk score, and they only mention reliability and no discussion of validity. Is there any reason why there would be bias in the magnitude of the coefficient because of what it is measuring that the other wasn't. Again, not necessary, just something to think about.

Thank you for this comment. We added a brief discussion of this point to the relevant paragraph in the Limitations section:

“A related limitation concerns the predictive strength of the polygenic scores available at the time of analysis. I rely on the educational attainment PGS from Lee et al. (2018), which, although foundational, has since been substantially improved through larger and more fine-grained GWAS efforts (e.g., Okbay et al., 2022). The use of an older, less predictive score introduces attenuation bias: weaker genetic instruments reduce the power to detect gene–environment interactions and likely make these estimates more conservative. This limitation primarily concerns reliability rather than construct validity, as newer PGSs capture the same underlying educational attainment phenotype with greater precision. Accordingly, I do not expect the use of an earlier score to bias coefficient magnitudes in a directional way, apart from classical attenuation due to measurement error. This bias works against finding significant effects, suggesting that the reported estimates may, if anything, understate the true magnitude of the underlying relationships.”

3. I would like to know why knowing the context is so important for these results matters. I very much agree with the statement in the conclusion, but I'd like to know why. This is very

important but there are a lot of things about this statement that could be interpreted differently and HOW would individual identities give us clues about the social moderation in that larger context as to the WHYs and the WHOs. Again, not a necessary statement, but it will resonate with a much larger readership when you give us more information about the 'so what.' I like what is included I simply wonder if a few more sentences can be added to truly show the significance of this type of work to scientific audience, writ large.

Thank you for this suggestion. We have added a brief discussion in the opening of the Discussion to clarify why understanding social context is central to interpreting these findings and to highlight the broader significance of this work for understanding heterogeneity in life-course outcomes and social mobility.

Thank you for your comments. My responses to your specific comments are below.

My original concerns have been addressed. In particular, the clarifications regarding the construction and interpretation of the PGS, the discussion of rGE and design limitations, and the more cautious framing of the power issues have substantially improved the manuscript. I do not have further major methodological concerns. I have only two comments left. First, I noticed that the title has been changed. While the revised version is more specific, I find that it overstates the strength of the evidence. The term “alter” implies a causal claim that goes beyond what the data can support. As the authors themselves acknowledge, several features of the study limit causal inference, including the between-family design and the modest statistical power of some key interaction terms. I would therefore suggest revising the title perhaps closer in tone to the original version.

Thank you for this comment. We agree that the revised title overstated the strength of the evidence. We have therefore revised the title to use explicitly associational language and to better reflect the limitations of the design. The revised title is: “Associations between genetic variants for educational success and risk and time preferences vary by childhood environment”

Second, although I did not raise this point in my first review, I recommend adding two to three references in the discussion that provide a general overview of polygenic score limitations (e.g., environmental confounding, population stratification, GWAS sample biases, assortative mating). While the authors already discuss several study-specific limitations, a brief general reference would help orient readers who are less familiar with the genetics literature.

Thank you for this suggestion. We now begin the Limitations section with references that provide a general overview of the conceptual and methodological challenges associated with polygenic scores, helping to orient readers who may be less familiar with the genetics literature.